# Inpainting the Sinogram from Computed Tomography using Latent Diffusion Model and Physics

## Abstract

Computed Tomography (CT) is a widely used non-invasive imaging technique for materials at microscopic or sub-microscopic length scales in synchrotron radiation facilities. Typically, the object is rotated relative to the X-ray beam, and 2D projection images are recorded by the detector at different rotation angles. The 3D object is then reconstructed by combining these projections and solving a computationally demanding inverse problem. The quality of the reconstructed image is critical for scientific analysis and is influenced by various factors, including the number of projections, exposure time or dose, and the reconstruction algorithm. In this work, we develop a foundation model by integrating a Generative AI-based Latent Diffusion Model (LDM) with physics-based domain knowledge. Specifically, we first develop and incorporate a set of loss functions into our LDM that accurately capture the physical properties of the CT data acquisition process. We demonstrate that adding these loss functions aids in stable training of the autoencoder in the LDM and improves its accuracy. The autoencoder and the diffusion model of the LDM are trained with real-world experimental data. Collecting real-world experimental data from synchrotron beamlines is often time-consuming and challenging. We demonstrate that the autoencoder trained with a combination of real-world experimental data and phantom shapes features also performs similarly to the autoencoder trained with real-world data. Second, we introduce a novel image blending method to combine the LDM's generated output with the original, extremely sparse sinogram data. Since our model integrates physics-guided loss functions focused on CT data acquisition, it simplifies the creation of downstream tasks and facilitates the adaptation of new features from different experiments. We demonstrate improvements of up to $23.5\%$ in SSIM for sinogram quality and $13.8\%$ for reconstructed image quality compared to state-of-the-art techniques.

## 1 Introduction

X-ray computed tomography (XCT) is a common and widely used non-invasive imaging technique at the synchrotron light sources (Sedigh Rahimabadi et al., 2020). XCT is used for many domain sciences, including imaging materials (Tang et al., 2021; Zhao et al., 2024; Intelligence Advanced Research Projects Activity), biological materials Keklikoglou et al. (2021), and others (Advanced Photon Source, Argonne National Laboratory). In a XCT experiment at the synchrotron beamlines, parallel X-ray beam is incident on an object placed on a rotation stage and rotated at different angles. For each rotation angle, the transmitted projection images are recorded in the detector based on the experimental geometrical configuration and alignment (Dyer et al., 2017), as shown in Fig. 1 . This is the focus use case in this work. Unlike medical/laboratory systems, synchrotron radiation facilities mostly use parallel beam geometry due to higher spatial and temporal resolution requirements. In medical/laboratory systems, typically the patient/object is stationary, with the source and detector rotating around the patient. We are not alluding to such medical experiments in this work.

The projection images can also be represented in terms of sinogram just by transposing the $x$ or $y$ coordinates of the projection image with that by the projection angle. Subsequently after all these projection images have been collected, a reconstruction algorithm is used that utilizes all these images to reconstruct the 3D object at high resolution. The quality of the reconstruction is essential

R4−Q1b−A
R4−Q1c−A

R2−W3−A
R2−Q3−A

R4−Q1d−A

for understanding the morphology and properties of the material and advancing scientific reasoning. Typically, to obtain high-quality 3D reconstruction, data is collected along a densely sampled tra-

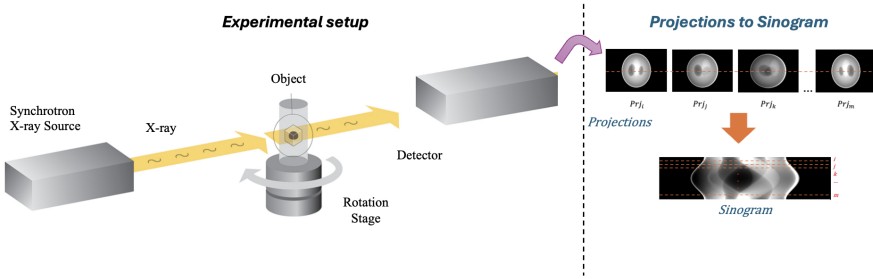

Figure 1: X-ray data acquisition schematic in a synchrotron beamline facility.

jectory during a CT experiment. However, this process is time-consuming – requiring hours or even days. Synchrotron radiation facilities provide high-energy X-rays that enable XCT experiments with high spatial and temporal resolutions. However, such high-energy beams also translate to a high ra-     R4−Q1e−A
diation dose on the sample, which can easily deform small features, especially when coupled with extended data acquisition times. In order to alleviate these issues , the data acquisition approach is     R4−Q1b−A
often modified, leading to sparse measurements. In one approach, the number of acquired projections is randomly reduced. In the second approach, the projection images are acquired sparsely at equal intervals, which leads to angular undersampling, referred to as the sparse view (SV) problem.     R4−Q1f−A
In a third approach, driven by geometric limitations of the rotation stage, there can be a range of angles where projections cannot be acquired - often referred to as the missing wedge or limited angle (LA) problem. Other incomplete projection data involves metal-corrupted projections, interior to-     R4−Q1f−A
mography problem and non-uniform detector problem (Wang et al., 2023). In this work, we address the SV and LA problems, where there are lack of projections results in band-like missing patterns in the sinogram, as shown in the input of stage-1 of the LDM in Fig. 2.

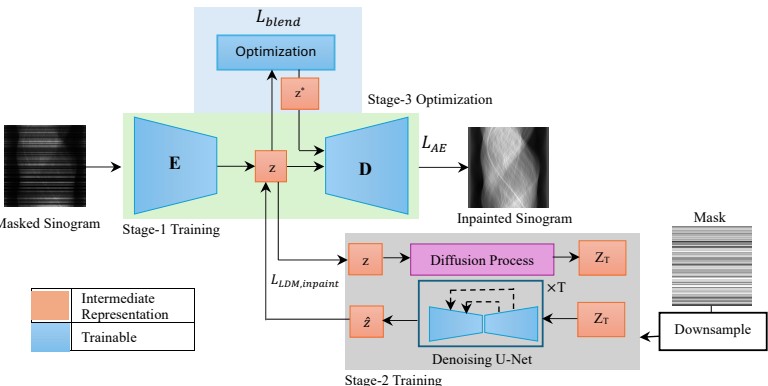

Figure 2: Overview of our algorithm. Stage-1 training: the autoencoder of the LDM is trained with a novel addition of physics-driven losses. Stage-2 training: the diffusion part of the LDM is trained. Stage-3: an optimization is done to blend the unmasked sinogram and the LDM prediction.     R3−Q2−A

In this paper, we improve the sparse data CT to dense data CT by inpainting the sinogram in the measurement domain, using our algorithm shown in Fig. 2. We develop a novel DL method with Generative AI-based LDM. The model is fed with missing sinogram data and the missing regions as masks. The LDM performs the task of sinogram inpainting and generates a sinogram with the filled information in the missing regions. The loss function of the autoencoder in the LDM is further improved with the domain-specific physics loss functions. In particular, we leverage the image formation process in the measurement and reconstructed object domains via the new loss function. Subsequently, we incorporate a novel blending algorithm to blend the output of the LDM with the

original sparse data. The entire model is trained and tested with real-world datasets and shows a significant improvement in image quality. Lastly, since our model is trained in the sinogram domain with a significant variety and volume of synchrotron radiation experimental data, it can be used as a foundation model for CT experiments and meet the requirements of many downstream ML tasks. We demonstrate this capability with LA and extremely sparse SV tasks. **Specifically, we make the following contributions: (1)** we develop a novel method of training the autoencoder in LDM by adding domain-specific physics knowledge of CT image formation for inpainting sinograms taking into account both measurement and reconstruction domains, which shows training stability of the autoencoder trained in an adversarial manner; **(2)** we develop a novel blending algorithm that improves the accuracy of inpainting tasks; **(3)** we develop two downstream tasks that address the SV and LA artifacts. Our downstream tasks use our foundation model which is trained with random masking; **(4)** we demonstrate the benefits of our approach for real-world tomographic data.

## 2 RELATED WORK

Various object reconstruction and analysis algorithms have been developed for limited CT data. The classical method of direct reconstruction from sinogram uses Filtered Backprojection (FBP) algorithm by Kak & Slaney (2001). However, FBP causes lot of artifacts in the CT reconstruction with limited projections. A Fourier grid reconstruction algorithm, Gridrec offers computational efficiency and less artifacts (Dowd et al., 1999; Rivers, 2012). Other approach is to address this ill-posed problem with iterative reconstruction algorithms such as Simultaneous Iterative Reconstruction Technique (SIRT) (Gilbert, 1972; Herman & Lent, 1976a;b; Van der Sluis & Van der Vorst, 1990), Simultaneous Algebraic Reconstruction Technique (SART) (Andersen & Kak, 1984), Discrete Algebraic Reconstruction Technique (DART) (Batenburg & Sijbers, 2011), Model-Based Iterative Reconstruction (MBIR)(Venkatakrishnan et al., 2014), Low-tilt Tomographic Reconstruction (LoTToR) (Zhai et al., 2020) and others. Although these methods can reduce image noise and artifacts, the reconstruction results are unsatisfactory without prior image information. A constrained total variation (TV) based iterative image reconstruction algorithm has also been developed (Sidky & Pan, 2008). Despite effectiveness of TV regularization in preserving edge and restoring smooth regions, fine features and image details are often overlooked. Another method of solving this problem using data-fidelity term with appropriate priors has been done (Kudo et al., 2013; Vandeghinste et al., 2011; Zhu et al., 2013). An alternative approach is to solve this problem as inpainting problem in the sinogram domain has the advantage of not suffering from aliasing artifacts.

Deep learning (DL)-based reconstruction algorithms have been popular in recent years, aiming to solve this limited CT problem. In this realm, GANs (Goodfellow et al., 2014; Valat et al., 2023) and U-Nets (Ronneberger et al., 2015) perform relatively well by considering the entire sinogram (Dong et al., 2019; Ghani & Karl, 2018; Tan et al., 2019; Yoo et al., 2019) or the local regions (Lee et al., 2018). One of the first to incorporate measurement and reconstruction domains by adding an inverse Radon transform layer is shown by Würfl et al. (2016). Approaches relevant to CT applications combining data from sinograms as well as reconstructed image domains are also used. An unsupervised sinogram inpainting network trained in both these domains has been shown by Zhao et al. (2018) for LA tomography. Similar principles of dual modalities in CT metal artifact reduction were presented by Lin et al. (2019) for removing metal artifacts and refining object reconstruction. The use of perceptual loss in networks has been presented in (Wei et al., 2020; Wu et al., 2020; Liu et al., 2020). A novel loss function and framework in both sinogram and image domain in a 2-step network for reconstruction from SV CT is presented by Wei et al. (2020). The use of local and global losses in the sinogram and residual error between reconstructed images has been presented by Yang et al. (2022). In Wu et al. (2021), reconstruction from SV CT has been done with the model consisting of embedding, refinement, and awareness modules. A similar approach by Ding et al. (2021) performs computationally efficient CT image reconstruction from SV CT in discrete Fourier domain. Adler & Öktem (2017) uses a gradient-like scheme while using prior information to solve the ill-posed inverse problem of CT reconstruction. Works such as Kofler et al. (2018) use a cascade of U-nets and data consistency layers to solve the SV CT problem, and the LA problem in CT as well (Yao et al., 2024). In recent years, Transformer (Dosovitskiy et al., 2020) architectures have been applied for the task of sparse CT upsampling. A Dense Residual Hierarchical Transformer network with attention-weighted loss is presented for sinogram inpainting by Adishesha et al. (2023). A transformer-based masked sinogram model for ill-posed problems is

R4−Q1d−A

R2−W5−A

R1−W1−A

R1−W1−A

R2−W4−A

addressed in Liu et al. (2022). All these DL methods often omit the consistency of measured data and result in the inaccurate representation of the image structure and features.

ADMM based DL approaches has been used for reconstructions from LA CT (Wang et al., 2019) and SV CT (Wang et al., 2022) which automatically adjusts the regularization. Such approaches are iterative algorithms with sensitivity to the training data. Combining iterative reconstruction with DL under the plug-and-play (PnP) framework have also been explored which adaptively learns the image priors to represent complex features and structures, while enhancing the reconstruction quality (Ye et al., 2018; Kamilov et al., 2023; He et al., 2019). PnP methods typically do not simultaneously consider local and nonlocal prior knowledge and is sensitive to the chosen denoiser. These DL based approaches (including ADMM and PnP) are often trained with limited features which limits its use for diverse applications. In E et al. (2024), a novel algorithm for inpainting of CT data based on LDM with the Fourier transform augmented autoencoder is presented. However, the work focuses on randomly masked projections, and does not demonstrate its feasibility to LA and SV tasks.

R2−W5−A

In general image inpainting tasks, a novel attention-based network (transformer) for image inpainting, based on an hourglass-shaped attention structure to generate appropriate features for complemented images is introduced by Deng et al. (2022). This paper also introduces Laplace attention based on Laplace distance prior for vanilla multi-attention head. A novel continuous-mask-aware transformer for image inpainting using masked attention and overlapping tokens is introduced by Ko & Kim (2023). A multi-level interactive Siamese filtering for inpainting of high-fidelity image inpainting has been proposed Li et al. (2022). To increase the receptive field in the inpainting network, Fourier Convolutions have been introduced Suvorov et al. (2022). Tackling semantic discrepancy in Diffusion Models for image inpainting to facilitate consistent and meaningful semantic generation has been introduced by Liu et al. (2024). CoPaint is introduced in Zhang et al. (2023), which can coherently inpaint the whole image without introducing mismatches. In Lugmayr et al. (2022), a novel algorithm RePaint is introduced. It uses the Denoising Diffusion Probabilistic Model (DDPM) for inpainting even in extreme masks. The vast majority of literature work focuses on RGB images, while our specific use case is based on grayscale sinogram images from XCT. Additionally, the features of interest in our experiments may be non-existent in most of the natural world datasets.

These works focus on generating or manipulating images globally, while editing images locally has received limited attention. In XCT experiments for synchrotron radiation beamlines, a small portion of whole projection is captured in a single exposure due to a limited field of view (Zhang et al., 2024), which needs to be stitched together to obtain a full projection image. In order to remove discontinuities and distortions during stitching, feature-based (Cheng et al., 2016) and cross-correlation based stitching methods (Vescovi et al., 2018) are used. However, these approaches limit the accuracy and stability of the stitched images. In XCT (especially medical applications), alpha image reconstruction (AIR) (Hofmann et al., 2014) approach is used which generates basis images based on certain properties (for example - high resolution, low noise), and subsequently generates voxel-specific weights which are applied to combine the basis images to have a final image with the desired properties. In medical Optical Coherence Tomography distortion corrections are also made with DL method (Qin et al., 2021). However, for synchrotron experiments, data types and features are diverse and data samples are limited. Blending of images, an approach which involves accurately fusing two images in local regions, has been aimed as well. Poisson image editing (Pérez et al., 2003) uses gradient driven reconstruction in pixels. Zhang et al. (2020) developed a differentiable model with Poisson loss, style loss, content loss and TV regularizer which improves the blending performance. Blended Diffusion (Avrahami et al., 2022) addresses zero-shot text-guided local image editing. Blending of outputs by optimizing the latent vector is demonstrated (Avrahami et al., 2023).

R1−W1−A

R4−Q2−A

## 3 METHODS

**3.1 PHYSICS-BASED LOSS FUNCTIONS FOR THE AUTOENCODER:** The inpainting task in this paper is to recover a dense sinogram $S_d \in \mathbb{R}^{P \times Dt}$ and the corresponding reconstructed image $I_d \in \mathbb{R}^{W \times H}$ from its counterpart missing data sinogram $S_s \in \mathbb{R}^{P \times Dt}$ and its corresponding reconstructed image $I_s \in \mathbb{R}^{W \times H}$. Here, $P$ is the number of projections, and $Dt$ is the number of detectors. $W$ and $H$ are the width and height of the reconstructed object respectively. Here, we are inspired by the LDM (Rombach et al., 2022), for the task of sinogram inpainting. The autoencoder consists of a combination of perceptual loss and physics-based loss objectives derived from the prin-

ciples of tomography data acquisition process. The encoder $\mathbf{E}$ takes a sparse sinogram $S_i \in \mathbb{R}^{P \times Dt}$ as input, and encodes it to a latent representation $z = \mathbf{E}(S_i)$. The sinogram $S_i$ denotes the $i$ th sinogram in the set of missing data sinogram $S_s$, thus, $S_i \subseteq S_s$. Here, the latent representation after encoding is $z \in \mathbb{R}^{w \times h \times c}$, with $w$ and $h$ as the width and height respectively, and $c$ is the channels of the latent representation $z$. The arbitrary high variances in the latent spaces is avoided by using the Vector Quantization (VQ) layer within the decoder $\mathbf{D}$. $z$ is quantized into $z_q$, and the backpropagation through the quantization operation is achieved using stop-gradient function $sg[.]$. The decoder $\mathbf{D}$    R2−Q2−A
reconstructs the image from the latent representation, $\tilde{S}_i = \mathbf{D}(z_q)$. The diffusion model works with the learned latent space representation, $z$. The training of the autoencoder through backpropagation follows a novel loss function, which consists of domain-specific physics penalty terms derived from the Tomographic data acquisition process. The original autoencoder loss function of Esser et al. (2021), shown in Eq. 1 consists of reconstruction loss term, perceptual loss term $L_P$, in addition, to    R1−Q1−A
the codebook loss (last two terms). Here $\beta = 0.25$.

$$L_{VQ}(\mathbf{E}, \mathbf{D}, Z) = |S_i - \tilde{S}_i| + L_P + ||sg[\mathbf{E}(S_i)] - z_q||_2^2 + \beta ||sg[z_q] - \mathbf{E}(S_i)||_2^2 \tag{1}$$

The autoencoder is trained in an adversarial manner using patch-based discriminator $Disc$ which differentiates the reconstructed sinogram from the actual sinogram. However, the loss formulation in Eq. 1 misses the underlying physical process of tomographic imaging. To capture the physical process of tomographic imaging, and prioritize the inpainting process for both the sinogram and object domains, we introduce additional novel physics-driven loss terms as enumerated below,

***(a) Hessian penalty for Sinogram:*** A sinogram is a sinusoidal wave-like pattern which is piecewise linearly continuous (Xie et al., 2017). The second-order derivatives of the sinogram provide the inflection points and peaks of these sinusoids. The second-order derivatives would be very sparse (Yang et al., 2022) and are incorporated by the Hessian penalty of the function (Boyd & Vandenberghe, 2004; Sun et al., 2015; Yang et al., 2022). To capture these sparse inflection points, the Hessian penalty term $L_H$ is introduced between reconstructed and ground truth sinograms, $\tilde{S}_i$, and $S_{i,gt}$ respectively, (Yang et al., 2022) in Eq. 2, with $\{a,b\} \in \{x,y\}$, and, $\frac{\partial^2}{\partial a \partial b} \in \{\frac{\partial^2}{\partial x^2}, \frac{\partial^2}{\partial x \partial y}, \frac{\partial^2}{\partial y \partial x}, \frac{\partial^2}{\partial y^2}\}$,

$$L_H = \sum_{x,y} \sqrt{Hs_{xx}^2 + Hs_{xy}^2 + Hs_{yx}^2 + Hs_{yy}^2} \text{with, } Hs_{ab} = \frac{\partial^2 \tilde{S}_i(a,b)}{\partial a \partial b} - \frac{\partial^2 S_{gt}(a,b)}{\partial a \partial b}. \tag{2}$$

***(b) Sinogram loss for Opposite Projections:*** A tomographic projection sums the transmitted X-rays passing through the object which is typically rotated around the object's central axis perpendicular to the direction of X-rays. Considering parallel X-rays with the detector placed at far field, the X-rays passing through the object rotated at an angle of $\alpha$ radians, follows identical trajectory as for the projection, with the object rotated at an angle of $\pi + \alpha$ radians (Yang et al., 2022). In sinogram domain, the sinogram at rotation angle $\alpha$ radians would be identical to the sinogram at rotation angle $\pi + \alpha$ radians, flipped around the detector's central axis. We utilize this property as loss function $L_O$ in the reconstructed sinogram as shown in Eq. 3,

$$L_O = \frac{\sum_{x,y} ||\tilde{S}_i(\alpha) - Fl_C(\tilde{S}_j)||_2^2}{P \times C}. \tag{3}$$

Here, we compute the mean error over the $P \times C$ pixels corresponding to the entire sinogram. $Fl_C(.)$ computes flipping of the sinogram with respect to the detector central axis, while we compute the difference of the reconstructed sinogram $\tilde{S}_i$ corresponding to rotation angle $\alpha$, and the reconstructed sinogram $\tilde{S}_j$ corresponding to rotation angle $\alpha + \pi$.

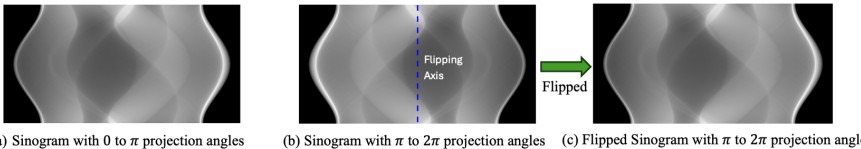

(a) Sinogram with 0 to $\pi$ projection angles    (b) Sinogram with $\pi$ to $2\pi$ projection angles    (c) Flipped Sinogram with $\pi$ to $2\pi$ projection angles

Figure 3: Identical sinograms for opposite projection angles. Sinogram with $\pi$ to $2\pi$ projection angles in (b) is flipped around the vertical axis to be transformed to (c), which is identical as (a).

*(c) Reconstruction loss:* We also match the reconstructed object by reconstructing the object from the output of the autoencoder ($\tilde{S}_i$) as well as the reconstructed object from the ground truth sinogram($S_{i,gt}$). The loss is formulated as $L_{RO}$, as shown in Eq. 4, where we compute the mean of the squared $L_2$ norm between the reconstructed object from the autoencoder ($\tilde{S}_i$) and ground truth ($S_{i,gt}$) using differentiable backprojection operator $FBP(.)$ in Eq. 4. Here $W$ and $H$ are the width and height of the reconstructed object. While performing sinogram to object reconstruction, a ramp filtering operation is required to remove the blurring effects .

R1$-$Q2$-$A

$$L_{RO} = \frac{\sum_{x,y} ||FBP(\tilde{S}_i) - FBP(S_{i,gt})||_2^2}{W \times H}. \tag{4}$$

*(d) Total Loss Function:* The total loss function $L_{AE}$ is the aggregation of the original loss function $L_{VQ}$, and the novel physics-driven domain losses. We weight each of these terms $L_H$, $L_O$, and, $L_{RO}$ by $k_1$, $k_2$, and, $k_3$ respectively as in Eq. 5. These weights are chosen in a heuristic approach, $k_1 = 10$, $k_2 = 10^3$, $k_3 = 10^5$ such that the contribution from each of these loss terms is equal. The autoencoder is trained in an adversarial manner for the sinogram inpainting task.

R3$-$W3$-$A
R4$-$Q6$-$A

$$L_{AE} = L_{VQ} + k_1 L_H + k_2 L_O + k_3 L_{RO} \tag{5}$$

Overall, the addition of these physics-based penalty terms introduces stability during the training of the Autoencoder. These losses also reduce the differences between the reconstructions and the ground truths in both the sinogram and reconstruction domains.

**3.2 DIFFUSION MODEL AND BLENDING:** The diffusion model of the LDM models the conditional distribution $p(z|y)$ with inputs $y$ as the condition. In this work, we use the masks as the conditional input $y$. The mask $y$ is downsampled and concatenated to the encoded masked sinogram, and mapped to intermediate layers of the U-Net using a cross-attention mechanism, which is defined as, $Attention(Q, K, V) = softmax(QK^T/\sqrt{d}).V$, with query $Q$, key $K$, and value $V$ containing trainable projection matrices. Based on the image-conditioning mask pairs, the conditional LDM is learned using the loss function in Eq. 6 with, $\epsilon_\theta$ is the neural backbone of the diffusion model, realized by U-Net model, and $z_t$ is the input latent representation of the $t$th equal sequence in the denoising U-Net, with $t = 1, ..., T$.

$$L_{LDM,inpaint} = \mathbb{E}_{\mathbf{E}(x),y\sim N(0,1),t}[||\epsilon - \epsilon_\theta(z_t, t, \tau_\theta(y))||_2^2]. \tag{6}$$

In the inpainting problem, given a missing data sinogram $S_s$ (Fig. 4(a)) and a binary mask $m$ corresponding to the missing regions, the LDM outputs a dense sinogram $\hat{S}_d$. Information from the sinograms $S_s$ and $\hat{S}_d$ is combined to form the final blended sinogram as $S_d = \hat{S}_d \odot m + S_s \odot (1 - m)$, as shown in Fig. 4(b), as the copy-paste sinogram, with $\odot$ being described as the element-wise product. The straightforward approach of combining the background and foreground objects by directly copying a foreground object from the source (inpainted) image and pasting it to the background object from the target image causes big intensity changes at the boundaries, creating artifacts visible to the human eye, as shown in Fig. 4(b). We blend the information from the foreground to that of the background in a novel approach, which minimizes the artifacts as illustrated in Fig. 4(c), compared to the actual sinogram shown in Fig. 4(d). We perform an optimization over the latent vector $z_0$ as a post-processing step to search for an optimal vector $z^*$ so that the masked area is similar to the edited image $\hat{S}_d$, and the unmasked region is similar to the input image $S_s$. This is shown in Eq. 7, where the sum of the mean squared error loss is computed between the edited image $\hat{S}_d$ and the decoder output for the masked region $m$, and the corresponding sparse image $S_s$ and the decoder output for the unmasked regions $1 - m$. The factor $\gamma$ preserves the fidelity to the background region $S_s$ ($\gamma = 1000$ used in our experiments).

$$L_{fid} = \frac{1}{N_{pixs}} \sum_{x,y} \{(\mathbf{D}(z) \odot m - \hat{S}_d \odot m)^2 + \gamma(\mathbf{D}(z) \odot (1 - m) - S_s \odot (1 - m))^2\} \tag{7}$$

Additional style loss and TV regularization are introduced in the optimization. The TV regularization loss drives the latent vector to remove the high-frequency noise. The style loss ($L_{style}$) aims to preserve the style of the input masked sinogram $S_s$ and the output sinogram $\hat{S}_{d1}$, as in Eq. 8.

$$L_{style} = \sum_{l=1}^{L} \frac{\beta_l}{2N_l^2} \sum_{i=1}^{N_l} \sum_{j=1}^{N_l} (G_l[S_d] - G_l[\mathbf{D}(z^*)]])_{ij}^2 \tag{8}$$

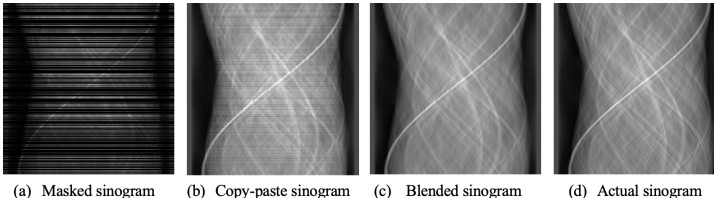

| (a) Masked sinogram | (b) Copy-paste sinogram | (c) Blended sinogram | (d) Actual sinogram |

Figure 4: Randomly masked sinogram and its inpainting approaches.

In Eq. 8, $\mathbf{D}$ is the decoder of the autoencoder in the LDM, $L$ is the number of convolutional layers, $N_l$ is the number of channels in activation, $M_l$ is the number of flattened activation values in each channel. $F_l[.] \in \mathbb{R}^{N_l \times M_l}$ is an activation matrix computed from a deep network $F$ at the $l^{th}$ layer. $G_l[.] = F_l[.]F_l[.]^T \in \mathbb{R}^{N_l \times N_l}$ denotes the Gram matrix of the corresponding activation at the $l^{th}$ layer which captures similarity relation between all pairs of channels that encode image style and texture. The weights $\beta_l$ control the influence of each layer. The TV loss ($L_{TV}$) is defined as Eq. 9.

$$L_{TV} = \sum_{m=1}^{H} \sum_{n=1}^{W} |\mathbf{D}(z^*)(m+1,n) - \mathbf{D}(z^*)(m,n)| + |\mathbf{D}(z^*)(m,n+1) - \mathbf{D}(z^*)(m,n)| \quad (9)$$

The total blending loss is defined as $L_{blend}$ in Eq. 10 with the multiplicative factors $p_{fid}$, $p_s$, $p_{TV}$ corresponding to the $L_{fid}$, $L_{style}$ and $L_{TV}$ respectively. In our experiments, $p_{fid} = 1$, $p_s = 10^4$, and, $p_{TV} = 10^{-6}$ is used such that the fidelity loss has higher contribution in the overall loss. The loss $L_{blend}$ is optimized for each image, with no separate training stage.

R4−Q6−A
R3−W3−A

$$L_{blend} = p_{fid}L_{fid} + p_s L_{style} + p_{TV}L_{TV} \quad (10)$$

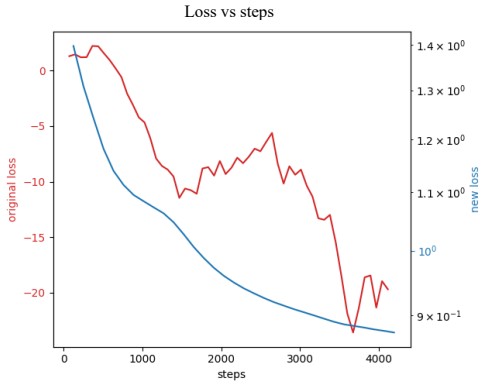

Table 1: Performance of **Autoencoder** trained with different loss settings

| AutoEncoder | Sinogram | | Reconstruction | |
|---|---|---|---|---|
| | SSIM | PSNR (dB) | SSIM | PSNR (dB) |
| Org loss | 0.9429 | 38.18 | 0.8571 | 35.64 |
| **New loss** | **0.9602** | **38.83** | **0.8944** | **37.42** |
| New loss w/o $L_O$ | 0.9590 | 38.84 | 0.8954 | 37.20 |
| New loss w/o $L_O$ and $L_H$ | 0.9544 | 37.79 | 0.8891 | 36.98 |

Table 2: Performance of **Autoencoder** trained with **New loss** for different training data

| Training Dataset | Sinogram | | Reconstruction | |
|---|---|---|---|---|
| | SSIM | PSNR (dB) | SSIM | PSNR (dB) |
| **Real World** | **0.9602** | **38.83** | **0.8944** | **37.42** |
| Real World + Phantom (Shapes) | 0.9400 | 33.95 | 0.8765 | 36.11 |
| Phantom (Shapes) | 0.6845 | 16.83 | 0.6290 | 29.16 |

Figure 5: Left: training loss over steps. Blue curve - autoencoder trained with new loss; red curve - autoencoder trained with the original loss. Table 1 shows the autoencoder performance at different loss settings . Table 2 shows the autoencoder performance trained with different data distributions.

R2−W1−A
R2−W2−A
R3−W2−A

## 4 EXPERIMENTAL RESULTS

The real-world data is curated from the XCT data present in TomoBank repository by De Carlo et al. (2018) based on feature complexity and contrast, alongwith the quality of sinogram and reconstruction images, which is termed as Exp_Data. We emphasize that collecting this real-world data from experiments at synchrotron beamlines is laborious and difficult to obtain. Sinogram data corresponding to the tomo_IDs of $1-4$, $23-26$, $31-56$, $64-75$, 77, 82, 85, 88, $90-93$, 96, 104, 107, 110 have been selected. More details about the dataset can be found in TomoBank webpage. The image resolution is different based on the corresponding experiments. In order to train and

R2−W2−A

R4−Q3−A

evaluate the DL model, we obtain a common image resolution size of $512 \times 512$ by performing data pre-processing in a step-wise manner. First, from the original projections, we reconstruct the object. Second, we reshape the object to a predefined shape and re-project the object to the desired rotation angles. Subsequently, the re-projections are converted to sinograms by transposing the projection angle axis with the $x$ axis of the projection image. This curated dataset falls in the realm of "small dataset", especially when compared to datasets with millions of images used for training foundation models, often used in literature. For training the autoencoder and diffusion model in the LDM, we use 50,000 training data, and 12,500 validation data randomly selected from Exp_Data. For comparing the autoencoder performance with different data distributions, the autoencoder is also trained with real-world data augmented with synthetic phantom data composed of simple shapes - circles, triangles, and polygons as shown in Table 2. The total training and validation data size is same in all the cases. The autoencoder is trained with the loss function $L_{AE}$ as shown in Eq. 5 for 4,000 steps. We use original unmasked sinograms to train the autoencoder, while for training the diffusion model, randomly masked sinograms with mask ratio varying between $0.1 - 0.9$ are used along with the binary masked regions during the pre-training stage. The pre-training of the diffusion model in the LDM has been done for 3,500 steps. For fine-tuning the diffusion model to different downstream tasks such as inpainting the SV and LA problems, the model utilizes fewer data, using 25,000 training data and 6,250 validation data selected randomly from Exp_Data. This fine-tuning do not require a high computational overload and can be implemented on a single compute node with upto 4 GPUs . We analyze the performance with 50 real-world test data samples that consist of simple and complex features. We used Polaris supercomputer and Lambda cluster at Argonne Leadership Computing Facility for model training and testing. Polaris supercomputer consists of 560 compute nodes, each having 4 NVIDIA A100 GPUs connected via NVLink. Lambda nodes, on the other hand, are DGX-1 machines that consist of 8 NVIDIA V100 GPUs each. We use Pytorch version 2.3.0 and CUDA version 12.4. Depending on the number of GPUs used during the training of the model, the learning rate ($lr$) can be defined as, $lr = lr_{base} \times grad\,accum\,steps \times GPU\,num \times batch\,size$. The training time for the autoencoder (stage-1) is 4 days using 6 NVIDIA V100 GPUs, while for pre-training the Diffusion Model (stage-2) takes 3 days, and, fine-tuning for the downstream tasks, the training time is 2 days - both using 4 NVIDIA V100 GPUs. The inference time for inpainting with the diffusion model is 9.23 seconds per image for 50 sampling steps in one NVIDIA V100 GPU. On the same hardware resource, the blending stage (stage - 3) is an iterative process that takes 0.69 seconds/iteration. Convergence of the blending algorithm is observed after 35 iteration steps.

R3−Q3−A

R1−Q3−A

R2−Q1−A

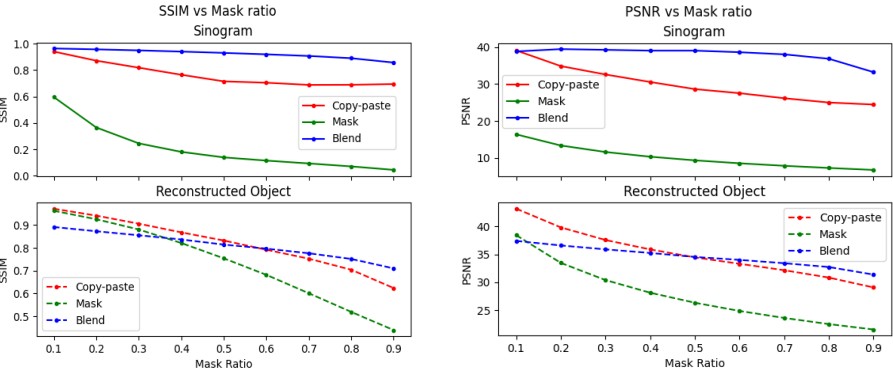

R1−W2−A
R3−W4−A

Figure 6: SSIM (left) and PSNR (right) vs mask ratio for sinogram (top) and reconstruction (bottom).

## 4.1 TRAINING AND PERFORMANCE OF AUTOENCODER

The autoencoder training is critical for the success of this model. We emphasize that training the autoencoder with additional physics losses aids in its stable training along with the improvement of the autoencoder's performance. Fig. 5 shows the training loss with and without the additional physics-based loss terms, which are the blue and red curves respectively. It is evident that the red curve oscillates a lot, while the blue curve converges smoothly during the training of the autoencoder. Table 1 shows the SSIM and PSNR performance of the autoencoder with various loss configura-

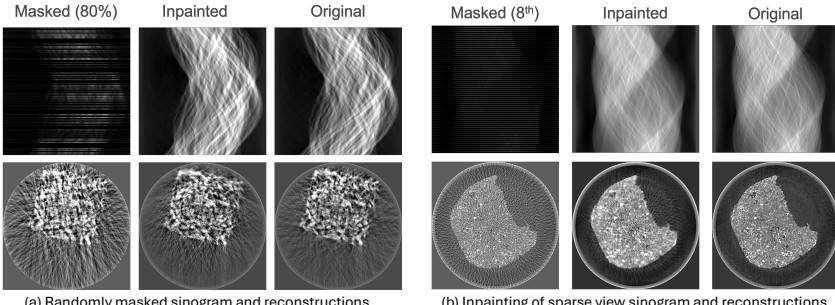

(a) Randomly masked sinogram and reconstructions    (b) Inpainting of sparse view sinogram and reconstructions

Figure 7: Inpainting of (a) $80\%$ randomly masked sinogram (top), and, (b) SV sinogram (every 8th projection acquired) (top), and its reconstructions (bottom).

R1−W2−A
R2−Q4−A
R4−Q4−A

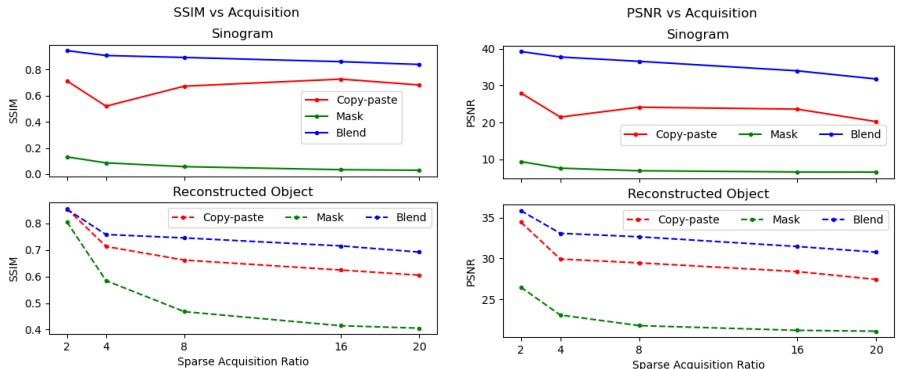

Figure 8: SSIM (left) and PSNR (right) vs SV acquisition for sinogram and reconstruction.

R1−W2−A

tions. We observe that the autoencoder trained with $L_{AE}$ performs best across the sinogram and reconstructed object domain. In Table 2, we demonstrate that combining the real-world data with synthetic data in $50 : 50$ ratio captures a wide range of features and performs close to the autoencoder trained with real-world data, while the one trained with only synthetic data performs worst.

## 4.2 PERFORMANCE FOR RANDOMLY MASKED DATA

Fig. 6 (left and right respectively) shows the variation of SSIM and PSNR over the mask ratio for the inpainted sinograms as well as for the reconstructed object obtained from these inpainted sinograms. In the sinogram domain (top row), the blending of the model's prediction with that of the unmasked sinogram indeed improves the performance, as seen by the blue and red solid plots. In the reconstructed object domain (bottom row), it is seen that the reconstruction from the sinogram with the masked region copied from the prediction and pasted to the unmasked region of the input sinogram produces better SSIM compared to the blended reconstructed object for lower mask ratios ($< 0.5$). This can be attributed to the TV loss introduced in the blending, which promotes smoother solutions. However, for higher mask ratios, the blended object performs better than the reconstruction from the copy-paste sinogram. For reconstructions from masked sinograms, it is observed that the SSIM is quite high for smaller mask ratios ($< 0.3$), which drops with the increase in mask ratio. With very sparse data (mask ratio $\sim 0.9$), the SSIM of reconstruction from the masked sinogram is the worst. The PSNR vs mask ratio plots (right column in Fig. 6) has similar trends as in the SSIM vs mask ratio plots (left column in Fig. 6). Fig. 7 (a) shows an example of real-world sinogram inpainting (top row) with $80\%$ of its data missing, and its reconstructions (bottom row). The PSNR values of the sinograms as well as the reconstructed object for the blended output surpass the copy-paste (sinogram and reconstructions) and mask reconstruction metrics. Overall, the trends of SSIM and PSNR over mask ratio are identical.

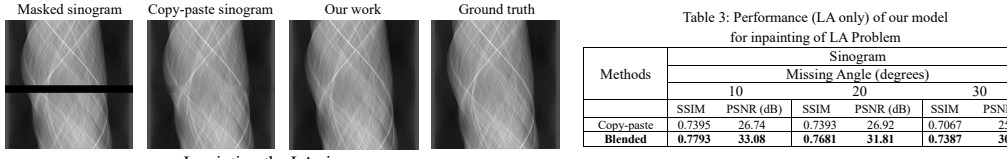

Figure 9: Left: Inpainting the LA sinogram. Table 3 shows performance for LA problem.

Table 3: Performance (LA only) of our model for inpainting of LA Problem

| Methods | Sinogram | | | | | |
| --- | --- | --- | --- | --- | --- | --- |
| | Missing Angle (degrees) | | | | | |
| | 10 | | 20 | | 30 | |
| | SSIM | PSNR (dB) | SSIM | PSNR (dB) | SSIM | PSNR (dB) |
| Copy-paste | 0.7395 | 26.74 | 0.7393 | 26.92 | 0.7067 | 25.47 |
| **Blended** | **0.7793** | **33.08** | **0.7681** | **31.81** | **0.7387** | **30.71** |

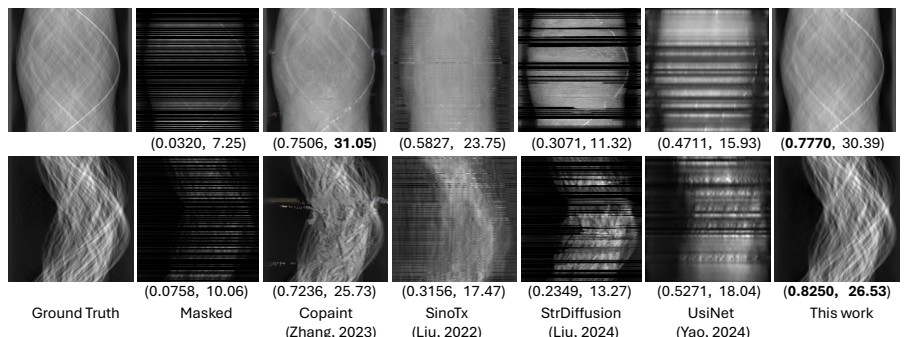

Figure 10: Inpainting performance comparison of our model with other state-of-art techniques for 80% random masking. (SSIM, PSNR) values provided at the bottom of each row.    R1−W2−A  R4−Q5−A

## 4.3 Performance for Downstream Tasks and Baseline Comparisons

***SV Acquisition:*** Inpainting in the realm of SV data is one of the downstream task after pre-training. Fig. 7 (b) shows the SV sinogram with the data acquired for every 8th sample and its inpainting (top row), and the reconstructed object (bottom row). Fig. 8 shows the inpainting of the SV data with the SSIM metrics in the sinogram (top left) and the reconstructed object domains (bottom left). The blended sinogram (blue solid) exceeds the copy-pasted sinogram (red solid) for all the sparse ratios. Additionally, the reconstructed object from the blended sinogram (blue dashed) plot surpasses the reconstructed object from the unmasked regions of the sinogram (green dashed plot), as well as the copy-pasted sinogram (red dashed plot). Identical trends are observed with the PSNR metrics as well as shown in Fig. 8(right). For all sparse ratios, the PSNR for the blended sinogram (blue solid), as well as the reconstructed object (blue dashed) from the blended sinogram, is maximum.

***Performance for LA Problem:*** Inpainting of the LA problem is another downstream task, which involves the inpainting of a bigger mask region. Fig. 9 (left) shows a sample performance of our model, while Table 3 shows the SSIM and PSNR metrics for inpainted and copy-paste sinogram. Clearly, the blended sinogram outperforms the copy-paste sinogram based on these metrics.

**Baseline Comparisons:** In order to obtain a fair comparison with other state-of-art algorithms, we compare our work with other inpainting approaches such as Copaint (Zhang et al., 2023), SinoTx (Liu et al., 2022), StrDiffusion (Liu et al., 2024), UsiNet (Yao et al., 2024) . Fig. 10 shows the comparative performance. We can see significant artifacts being present in inpainting from SinoTx and Copaint. StrDiffusion and UsiNet contains horizontal dark stripes which is erroneous. We demonstrate our model as the best performing one with CoPaint performing closely, but with artifacts.    R1−W1−A

## 5 Conclusion

This paper presents a novel method combining domain-specific physics knowledge with the SDM for inpainting in the sinogram domain from real-world CT experimental data. It introduces the physics loss functions in the sinogram as well as in the reconstructed object domains and a novel blending algorithm. It develops a foundation model pre-trained on random masking and fine tunes it on tasks such as sparsely acquired data and missing wedge problems. Our model outperforms the state-of-the-art baselines by $23.5\%$ for sinogram and $13.8\%$ reconstructed object in terms of SSIM.

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
