# OpenReview forum: "Inpainting the Sinogram from Computed Tomography using Latent Diffusion Model and Physics"
_ICLR.cc/2025/Conference — Submitted to ICLR 2025_

### Official Review · Reviewer_jFB8 · 2024-11-02

**Soundness:** 2
**Presentation:** 1
**Contribution:** 2
**Rating:** 1
**Confidence:** 5

**Summary:**

This paper proposes a method that integrates a Latent Diffusion Model (LDM) with physics-based loss functions to address missing data issues in CT imaging by inpainting sinograms. The paper demonstrates an approach by combining machine learning techniques with CT-specific physical properties. However, the manuscript has shortcomings regarding organization, precision, and detail.

**Strengths:**

This paper presents an approach by integrating a Latent Diffusion Model (LDM) with physics-based loss functions for CT sinogram inpainting, demonstrating originality in combining generative AI with domain-specific knowledge. The work introduces unique physics-driven loss functions that enhance the stability and accuracy of the model.

**Weaknesses:**

The manuscript is challenging to read due to disorganized logic, unprofessional expressions, and vague descriptions, making it hard to follow the motivation and methodology.

**Questions:**

1.     This manuscript is challenging to read, with disorganized logic and non-standard, unprofessional expressions:
	•	The paper only lists existing methods without clearly identifying their limitations or explaining the motivation for the proposed approach.
	•	Expressions such as “During a CT experiment” and “CT experimental time” are imprecise and lack professionalism.
	•	The statement, “The object is rotated stepwise around a central axis,” is misleading; typically, it is the source and detector that rotate, not the object itself.
	•	Phrases like “Combining several such projections, a specialized algorithm” are vague and difficult to understand. Please clarify.
	•	The sentence, “it can also lead to morphological deformation in the sample due to extended radiation exposure,” seems to imply that deformation results from patient motion over extended scan times.
	•	The phrases “In one approach,” “In another approach,” and “In a third approach” might be referring to sparse-view CT, limited-angle CT, etc. However, other forms of incomplete projection data also exist, which could be covered or referenced in a comprehensive review article [1].

[1] Wang T, Xia W, Lu J, et al. A review of deep learning ct reconstruction from incomplete projection data. IEEE Transactions on Radiation and Plasma Medical Sciences, 2023.

2.     The so-called “novel blending algorithm” proposed in this paper is, in fact, a common practice in CT reconstruction from incomplete measurements.

3.     The term “realistic simulated data” raises ambiguity regarding the authenticity of the experimental data; it is unclear whether the data are from real experiments or simulations. Although the paper emphasizes the importance of physics-based loss functions, there is no detailed description of imaging parameters. Additionally, the use of “re-project” suggests that the projection data may not be genuine, but rather re-generated, which further questions the authenticity of the data used.

4.     Since the method uses FBP with a reconstruction loss, why not present the reconstructed CT images? This would provide a more intuitive understanding of the results.

5.     The resolution of Figure 9 is too low, making it difficult to read the results.

6.     The paper involves many hyperparameters but lacks any analysis or justification regarding their selection. Given the importance of these parameters to the model’s performance, a sensitivity analysis or at least a discussion on hyperparameter tuning would enhance the reliability and reproducibility of the results.

---

> ### Author Response · Authors · 2024-11-22
> **Reviewer jFB8 (Reviewer 4) comment summary : We are grateful to the reviewer for the valuable time, insightful comments and constructive suggestions. We address the weaknesses and answer the questions. We made changes in the revised paper with blue and tag our response.**
>
> Answers:
>
> R4-W1-A: The manuscript has been modified to improve upon the logic, expressions and descriptions.
>
> R4-Q1a-A: Added in Section – 1 (Introduction) and Section – 2 (Related Work) of the revised manuscript.
>
> R4-Q1b-A: These expressions have been removed and rephrased in the revised manuscript. Please see line 44 and 74 of the revised manuscript.
>
> R4-Q1c-A: In the experiments at synchrotron beamlines, the object is placed on a rotation stage and rotated at different angles. For each rotation angle, the parallel X-ray beam is incident on it, and the transmitted X-ray projection images are recorded in the detector based on the experimental geometrical configuration and alignment. This is the use case of focus in this work. On the contrary, in medical/laboratory CT experiments, typically the patient/object is stationary, with the source and detector rotating around the patient. We are not alluding to such medical experiments in this work. Please see line 45 of the revised manuscript.
>
> R4-Q1d-A: The 2D projection images are recorded in the detector for the object rotated at different rotation angles. Subsequently after all these projection images has been collected, a reconstruction algorithm is used which utilizes all these images to reconstruct the object. Please see line 52-53 and 124-133 of the revised manuscript.
>
> R4-Q1e-A: Synchrotron radiation facilities provide high-energy X-rays that enable XCT experiments with high spatial and temporal resolutions. However, such high-energy beams also translate to the high radiation dose on the sample, which can easily deform small features, especially when coupled with extended data acquisition times. We are not referring to the deformation caused by the patient motion over extended scan times. Please see line 71-74 of the revised manuscript.
>
> R4-Q1f-A: In the revised manuscript we highlighted the other incomplete projection data problems and referenced the above review paper. Although, these can be implemented as other downstream AI/ML tasks using our model, we focus on sparse-view CT, limited-angle CT use cases. Please see line 76 and lines 78-81 of the revised manuscript.
>
> R4-Q2-A: In XCT experiments pertaining to synchrotron radiation beamlines, there can be small portion of whole 2D projection captured in single exposure due to limited field of view.  The scanned region needs to be stitched together to obtain full-projection image.  In order to remove discontinuities and distortions during stitching, feature-based and cross-correlation based stitching methods are used. However, these approaches limit the accuracy and stability of the stitched images. In XCT (especially medical applications), alpha image reconstruction (AIR) approach is used which generates basis images based on certain properties (high resolution, low noise, etc.), and subsequently generates voxel-specific weights to have a final image with the desired properties. In medical Optical Coherence Tomography (OCT) corrections are also made with DL methods. However, in synchrotron beamline experiments, the data types and features are diverse and data samples are limited. Our novel blending algorithm for CT optimizes the latent vector from the latent diffusion model (LDM) to match the fidelity of the original unmasked sinogram data with that of the LDM output. This has been added in lines 189 – 202 of the revised manuscript.
>
> R4-Q3-A: The real-world data has been curated from the real experimental data in the TomoBank databank. We select the data based on feature complexity and contrast, alongwith the quality of sinogram and reconstruction images. The imaging parameters for each experimental data are different. The details can be found in the Tomopy paper, https://iopscience.iop.org/article/10.1088/1361-6501/aa9c19, and https://tomobank.readthedocs.io/en/latest/source/data.html. TomoBank is a collection of CT data from different experiments at the synchrotron beamlines. These data are collected at different scan ranges and projections. To bring them to a uniform data size, we perform the operation of reconstruction followed by re-projection. The authenticity of the data is maintained in this process. This has been added in lines 372-377 of the revised manuscript.
>
> R4-Q4-A: The reconstructed images are being presented in Fig. 7 of the revised manuscript. Please see lines 443-444 of the revised manuscript.
>
> R4-Q5-A: Figure 9 (now figure 10) has been updated in the revised manuscript. Please see lines 507-508 of the revised manuscript.
>
> R4-Q6-A: In this work, the hyperparameters are chosen based on a heuristic approach. For example, in Eqn. 5, the multiplicative factors k1, k2, and k3, are chosen such that the contribution of each of these loss terms is equal. In the blending step in Eqn. 10, the hyperparameters are chosen such that the contribution of the fidelity loss is higher in the overall loss. This has been added in line 283 and line 346 of the revised manuscript.

---

> > ### Comment · Reviewer_jFB8 · 2024-11-27
> >
> > Based on the efforts made by the authors during the rebuttal phase, I am willing to raise my score from 1 to 3. This adjustment is more a recognition of the authors’ dedication than an indication of improved manuscript quality. I still believe that this work is not suitable for acceptance in its current form and recommend that the authors further refine their methodology, enhance experimental validation, and deepen theoretical analysis before considering resubmission.

---

> > > ### Author Response · Authors · 2024-11-27
> > >
> > > We thank the Reviewer jFB8 for the additional comments. It would be really helpful, if the reviewer can let us know in what specific way the revised manuscript can be improved in terms of methodology, experimental validation, and theoretical analysis?
> > > To the best of our knowledge, we have answered all the concerns which were raised by the Reviewer jFB8.

---

### Official Review · Reviewer_2Asj · 2024-11-02

**Soundness:** 3
**Presentation:** 3
**Contribution:** 3
**Rating:** 6
**Confidence:** 3

**Summary:**

This paper deals with the problem of sparse sinograms in CT. The authors combine an autoencoder with a latent diffusion model which incorporates physics knowledge in the loss functions. This stabilizes the training process of the autoencoder. Furthermore, a blending algorithm is applied such that the output of the diffusion model also aligns with the measurement data.

**Strengths:**

* Including physics-domain knowledge is an essential task in machine learning.
* Sparse sinograms are problems in practice to reduce the dose and/or the acquisition time.
* The method was tested on a real-world dataset.
* The paper has a good structure, explaining the individual components.

**Weaknesses:**

* The proposed method seems very complex. There are many different sub-modules: autoencoder, diffusion model, blending.
* Training an autoencoder is hard.
* There are many hyperparameters to adjust in the different loss functions.
* The blending technique is not suitable for lower mask ratios.

**Questions:**

* Do we really need such a complex setting? Isn’t it possible to use a simpler model like a diffusion model and include all the losses there?
* Is this approach end-to-end trainable?
* What are phantom features?

---

> ### Author Response · Authors · 2024-11-22
> **Reviewer 2Asj (Reviewer 3) comment summary : We are thankful to the reviewer for the valuable time, insightful comments and constructive suggestions. We address the weaknesses and answer the questions. We made changes in the revised paper with blue and tag our response.**
>
> Weaknesses:
>
> Weakness 1 (R3-W1): The proposed method seems very complex. There are many different sub-modules: autoencoder, diffusion model, blending.
>
> Answer (R3-W1-A): Although the proposed method seems to be very complex, this approach provides computational efficiency, high-quality image generation, fine-grained control, improved training stability, scalability and easy adaptation to other tasks compared to other techniques.
>
> Weakness 2 (R3-W2): Training an autoencoder is hard.
>
> Answer (R3-W2-A): While training the autoencoder is hard, the addition of physics driven loss functions provides training stability as shown in Fig. 1 in the paper. Please see line 367 of the revised manuscript.
>
> Weakness 3 (R3-W3): There are many hyperparameters to adjust in the different loss functions.
>
> Answer (R3-W3-A): The hyperparameters in the different loss functions have been optimized heuristically which provides state-of-art results for CT Sinogram inpainting tasks. For the autoencoder, the goal is to have the loss contribution from different terms equal, while for the blending stage, higher priority was given to fidelity loss. Further hyperparameter selection and optimization can be done in future research directions. Please see line 283 and 346 of the revised manuscript.
>
> Weakness 4 (R3-W4): The blending technique is not suitable for lower mask ratios.
>
> Answer (R3-W4-A): The blending technique shows significant improvement in the sinogram inpainting compared to the copy-paste approach for all the mask ratios. However, for the reconstructed object, the SSIM and PSNR improvements are not significant in lower mask ratios, due to TV regularization dominating at lower mask ratios. All these results shown in Fig. 6 of the paper. We aim to improve on the reconstruction results in future research work. Please see line 422 of the revised manuscript.
>
> Questions:
>
> Question 1 (R3-Q1): Do we really need such a complex setting? Isn’t it possible to use a simpler model like a diffusion model and include all the losses there?
>
> Answer (R3-Q1-A): In this paper, we focus on developing the sinogram inpainting using LDM due to its higher computational efficiency, flexibility, and preserving fine image details compared to standard diffusion models. Beamlines at synchrotron radiation facilities image wide range of samples/specimens from different scientific domains, resulting in varying types of features of interests (e.g., integrated circuit and mouse brain). Therefore, it is important to provide efficient training (and / or fine tuning) capabilities for down stream AI/ML tasks. Nevertheless, the feasibility of developing a physics-informed diffusion model for sinogram inpainting tasks and its comparison with current work is one of the future research endeavors.
>
> Question 2 (R3-Q2): Is this approach end-to-end trainable?
>
> Answer (R3-Q2-A): The algorithm is trained in stages. In the first stage, the autoencoder is trained with novel physics-driven losses. In the second stage, the diffusion part is trained. This training strategy has been adopted from the approach of the original LDM paper (https://openaccess.thecvf.com/content/CVPR2022/html/Rombach_High-Resolution_Image_Synthesis_With_Latent_Diffusion_Models_CVPR_2022_paper.html), and results in better performance in the scenario where such a model (LSGM) is trained end-to-end (https://nvlabs.github.io/LSGM/). The blending of the unmasked sinogram and the LDM output involves no training and is optimized during the inference stage for each sinogram image. During the inference stage, the algorithm is executed end-to-end. Please see line 99 of the revised manuscript.
>
> Question 3 (R3-Q3): What are phantom features?
>
> Answer (R3-Q3-A): The phantom features involve synthetic objects generated in Python composed of simple shapes such as circles, triangles, and other polygons. Although, these features do not represent real-world features, their sinograms can be used for training as we show with our experimental evaluation. Please see line 385-389 of the revised manuscript.

---

### Official Review · Reviewer_tPkr · 2024-11-03

**Soundness:** 3
**Presentation:** 3
**Contribution:** 2
**Rating:** 3
**Confidence:** 5

**Summary:**

The paper proposed a model which integrate the Latent Diffusion Model (LDM) with physics-based domain knowledge. A set of loss functions were designed. The paper proposed a blending algorithm to improve the accuracy of inpainting task. The proposed method is work for simulated parallel projection data (not real-world data).

**Strengths:**

(1)	domain specific physics knowledge of CT image formation for inpainting sinograms taking into account both measurement and reconstruction domains.
(2)	Recover sinogram with different masks and different sampling ratio.
(3)	Suitable for Sparse data acquisition and Missing Wedge Problem.

**Weaknesses:**

(1)	The loss is complex and too many parameters. Ablation of every part is necessary.
(2)	The paper uses the simulated projection data other than real word data.
(3)	The proposed method is only work with Parallel beam projection geometry with is xxxxx
in real application.

(4)	The downstream task of the method is image reconstruction. Comparison with reconstruction method for sparse view reconstruction and limit view reconstruction is necessary, such as dual domain reconstruction.
[1] W. Wu, D. Hu, C. Niu, H. Yu, V. Vardhanabhuti and G. Wang, "DRONE: Dual-Domain Residual-based Optimization NEtwork for Sparse-View CT Reconstruction," in IEEE Transactions on Medical Imaging, vol. 40, no. 11, pp. 3002-3014, Nov. 2021, doi: 10.1109/TMI.2021.3078067
[2] Ding, Q., Ji, H., Gao, H., Zhang, X. (2021). Learnable Multi-scale Fourier Interpolation for Sparse View CT Image Reconstruction.
(5) The literature review is limited, many CT reconstruction works, such as iterative reconstruction and deep learning reconstruction (image domain, unrolling (ADMM-Net), and plug-&play method) are not given

**Questions:**

1 The training and inference time comparison is necessary for sampling is time-consuming.
2 Notation is not defined clearly, e.g. equation (1) sg. and z_qn never used.
3 Can the method extend to 2D fan-beam and 3D Cone beam reconstruction
4 Visual comparison of reconstructed image from recovered sinogram also needed for that a minor error of sinogram may lead to streaky artifacts in image.

---

> ### Author Response · Authors · 2024-11-22
> **Reviewer tPkr (Reviewer 2) comment summary : We thank the reviewer for the valuable time, insightful comments and constructive suggestions. We address the weaknesses and answer the questions. We made changes in the revised paper with blue and tag our response.**
>
> Weaknesses:
> Weakness 1 (R2-W1): The loss is complex and too many parameters. Ablation of every part is necessary.
>
> Answer (R2-W1-A): The ablation study of the autoencoder loss considering different loss terms has been provided in Table – 1. Overall, adding all the physics terms aids in stable training and improved performance compared to the original LDM as shown in Fig. 5. Please see line 365 of the revised manuscript.
>
> Weakness 2 (R2-W2): The paper uses the simulated projection data other than real word data.
>
> Answer (R2-W2-A): We would like to clarify that the paper uses real-world data as described in Section 4 (experimental results) for training and inference of the models. The dataset is extracted from TomoBank database. It contains more complex data than the simulated shapes data, due to the presence of more complex features derived from the real-world data. Please see line 372 of the revised manuscript. Additionally, in Table – 2, we show that mixing this real-world data with simulated shapes data in 50:50 during training the autoencoder results in its peformance close to	that achieved with only the real-world data. Please see line 366 of the revised manuscript.
>
> Weakness 3 (R2-W3): The proposed method is only work with Parallel beam projection geometry with is xxxxx in real application.
>
> Answer (R2-W3-A): The proposed method has been developed specifically for X-ray Computed Tomography 	(XCT) experiments in the synchrotron beamlines. Unlike laboratory systems, synchrotron radiation facilities mostly use parallel beam geometry during XCT data acquisition due to the higher spatial and temporal resolution requirements. Please see line 47 of the revised manuscript.
>
> Weakness 4 (R2-W4): The downstream task of the method is image reconstruction. Comparison with reconstruction method for sparse view reconstruction and limit view reconstruction is necessary, such as dual domain reconstruction.
>
> Answer (R2-W4-A): The current work focuses on sparse data acquisition and missing wedge as AI/ML downstream tasks. We use FBP to reconstruct the outputs of our AI/ML downstream tasks. We will investigate reconstruction as an AI/ML task in the future.  The reference to these two papers has been made and discussed in the revised manuscript. Please see lines 153-156 of the revised manuscript.
>
> Weakness 5 (R2-W5): The literature review is limited, many CT reconstruction works, such as iterative reconstruction and deep learning reconstruction (image domain, unrolling (ADMM-Net), and plug-&play method) are not given
>
> Answer (R2-W5-A): In the revised manuscript, works such as iterative reconstruction and deep learning reconstruction (image domain, unrolling (ADMM-Net), and plug-&-play methods) has been provided. Please see lines 128 – 133 of the revised manuscript for the iterative reconstruction methods, and lines 164-172 of the revised manuscript for Deep learning reconstruction methods.
>
> Questions:
> Question 1 (R2-Q1): The training and inference time comparison is necessary for sampling is time-consuming.
>
> Answer (R2-Q1-A): The training time for the autoencoder (stage-1) is 4 days using 6 NVIDIA V100 GPUs. The training time for pre-training the Diffusion Model (stage-2) is 3 days, while for fine-tuning for the downstream tasks, the training time is 2 days - both using 4 NVIDIA V100 GPUs. The inference time for inpainting with the diffusion model is 9.23 seconds per image for 50 sampling steps in one NVIDIA V100 GPU. On the same hardware resource, the blending stage (stage - 3) is an iterative process that takes 0.69 seconds/iteration. 	Convergence of the blending algorithm is observed after 35 iteration steps. Please see lines 402-407 of the revised manuscript.
>
> Question 2 (R2-Q2): Notation is not defined clearly, e.g. equation (1) sg. and z_qn never used.
>
> Answer (R2-Q2-A): We expanded Section 3.1 of the paper and included a more detailed and descriptive 	narration of the notations. The explanations and use of sg. and z_q is also provided. Please see lines 221-222 of the revised manuscript.
>
> Question 3 (R2-Q3): Can the method extend to 2D fan-beam and 3D Cone beam reconstruction
>
> Answer (R2-Q3-A): Yes, the method can easily be extended to 2D fan-beam and 3D Cone beam reconstruction. However, the physical loss for the transformation between the projection/sinogram space to the reconstructed object space needs to be taken into account as well. In this work, considering our specific use case of synchrotron beamline experiments, we 	limit our study to 2D parallel beam X-rays only. Please see lines 48-50 of the revised manuscript.
>
> Question 4 (R2-Q4): Visual comparison of reconstructed image from recovered sinogram also needed for that a minor error of sinogram may lead to streaky artifacts in image.
>
> Answer (R2-Q4-A): We included the reconstructed images in Fig. 7 of the revised manuscript.  Please see line 443 of the revised manuscript.

---

> > ### Comment · Reviewer_tPkr · 2024-11-27
> >
> > Your responses cannot addressed the majority of my concerns. After carefully considering both the original content of the paper and your clarifications in the reply, I have decided to maintain my initial scoring.

---

> > > ### Author Response · Authors · 2024-11-27
> > >
> > > It would be really good if the Reviewer tPkr can point out specific concerns which has not been addressed in the revised paper and in the comment. This would help us in making the paper better.
> > > Current feedback from the Reviewer is too vague.

---

> > > > ### Comment · Reviewer_tPkr · 2024-11-28
> > > >
> > > > 1、Comparison with dual domain method is necessary for this method focus mainly on sinogram.
> > > > 2、In this version, the author said the method limited to parallel projection. However, I want the evaluation of the method with fan beam projection from the former version.
> > > > 3、The comparison of computation time with the other methods is not given.
> > > > I still believe that this work is not suitable for acceptance.

---

### Official Review · Reviewer_4qQL · 2024-11-03

**Soundness:** 3
**Presentation:** 1
**Contribution:** 2
**Rating:** 3
**Confidence:** 4

**Summary:**

This paper presents a novel foundation model for sinogram inpainting based on the latent diffusion model (LDM). By considering certain physical characteristics of CT acquisition and sinograms, they designed an innovative physics-informed loss function that significantly enhances the autoencoder's training performance. Additionally, they introduced a sinogram blending technique that balances content and style perception to improve the synthesis quality between predicted and real sinograms. Experimental results demonstrate that this method outperforms current state-of-the-art approaches in sinogram inpainting, highlighting its potential as a foundational model for CT applications.

**Strengths:**

By incorporating the physical characteristics of CT acquisition and sinograms into the design of the loss function, this approach effectively combines domain knowledge with advanced techniques, enabling more efficient learning for domain-specific tasks.

**Weaknesses:**

- The structure of this paper is highly disorganized. For instance, in lines 171–176, after introducing various works on sinogram inpainting, the authors abruptly shift to discussing image blending, which may confuse readers. Furthermore, the authors seem unfamiliar with the proper use of `\citet` and `\citep` commands, as nearly all citations fail to meet academic writing standards, particularly in Section 2, where incorrect citation formatting severely disrupts the reading experience. In Section 4.4, almost all citations are wrong, with many references not corresponding to their intended sources. The authors should thoroughly review and correct all citations with careful attention to detail.
- Figure 9 is severely blurred, making it difficult to discern the comparative effectiveness of the proposed method against other approaches. Additionally, the paper lacks a quantitative performance comparison with the baseline methods. Although the primary task is sinogram inpainting, in practical applications, the focus is ultimately on the quality of the final CT images reconstructed from the inpainted sinograms. This aspect is missing from the current work. The authors should consider including both quantitative and qualitative results of the CT images reconstructed from sinograms completed using the proposed method.

**Questions:**

- In line 206, the phrase “perceptual loss term, $L_P$ in addition” seems to have incorrect comma placement. It should be written as “perceptual loss term $L_P$, in addition,” correct?
- In line 263, the authors state, “we also need a ramp filtering operation for noise removal in the image.” This is inaccurate; it should be clarified that the purpose of the ramp filtering operation is to remove blurring effects.
- In line 367, the authors mention, “we fine-tune the model with fewer data,” and that “this fine-tuning approach does not require a high computational overload and can be implemented on a compute node with limited GPU resources.” However, they also mention using 25,000 real-world training samples for fine-tuning, which is not a small dataset. This raises the question of how it can be feasible “with limited GPU resources.”

---

> ### Author Response · Authors · 2024-11-22
> **Reviewer 4qQL (Reviewer 1) comment summary : We appreciate the reviewer for the valuable time, insightful comments and constructive suggestions. We address the weaknesses and answer the thoughtful questions. We made changes in the revised paper with blue and tag our response.**
>
> Weaknesses:
>
> Weakness 1 (R1-W1): The structure of this paper is highly disorganized. For instance, in lines 171–176, after introducing various works on sinogram inpainting, the authors abruptly shift to discussing image blending, which may confuse readers. Furthermore, the authors seem unfamiliar with the proper use of \citet and \citep commands, as nearly all citations fail to meet academic writing standards, particularly in Section 2, where incorrect citation formatting severely disrupts the reading experience. In Section 4.4, almost all citations are wrong, with many references not corresponding to their intended sources. The authors should thoroughly review and correct all citations with careful attention to detail.
>
> Answer (R1-W1-A):
>             The manuscript has been revised to provide better transition between various works on sinogram inpainting and image blending.
> The transition text has been added in line 187 – 191. \citet and \citep commands has been used to reformat the citations in the revised manuscript. In Sections 2 and 4.4, the citations have been edited in the revised manuscript. The text has been added in line 133 and 138 as examples. However, changes has been made to the entire revised manuscript. Citations has been modified in line 528 of the revised manuscript as well.
>
> Weakness 2 (R1-W2): Figure 9 is severely blurred, making it difficult to discern the comparative effectiveness of the proposed method against other approaches. Additionally, the paper lacks a quantitative performance comparison with the baseline methods. Although the primary task is sinogram inpainting, in practical applications, the focus is ultimately on the quality of the final CT images reconstructed from the inpainted sinograms. This aspect is missing from the current work. The authors should consider including both quantitative and qualitative results of the CT images reconstructed from sinograms completed using the proposed method.
>
> Answer (R1-W2-A): Fig. 9 has been improved with the quantitative metrics provided with the baseline methods. The reconstructions have also been added in the revised manuscript in Fig. 7 with the quantitative metrics provided in bottom rows of Figs. 6 and 8. Please see line 507, 442, 420, 459 of the revised manuscript.
>
>
> Questions:
>
> Question 1 (R1-Q1): In line 206, the phrase “perceptual loss term, LP in addition” seems to have incorrect comma placement. It should be written as “perceptual loss term LP, in addition,” correct?
>
> Answer (R1-Q1-A): Corrected in the revised manuscript. Please see line 227 of the revised manuscript.
>
> Question 2 (R1-Q2): In line 263, the authors state, “we also need a ramp filtering operation for noise removal 	in the image.” This is inaccurate; it should be clarified that the purpose of the ramp filtering operation is to remove blurring effects.
>
> Answer (R1-Q2-A): We thank the reviewer for pointing out the mistake. Yes, the purpose of the ramp filtering operation is to remove blurring effects during the object reconstruction. We corrected the sentence in the revised manuscript. Please see line 276 of the revised manuscript.
>
> Question 3 (R1-Q3): In line 367, the authors mention, “we fine-tune the model with fewer data,” and that “this fine-tuning approach does not require a high computational overload and can be implemented on a compute node with limited GPU resources.” However, they also mention using 25,000 real-world training samples for fine-tuning, which is not a small dataset. This raises the question of how it can be feasible “with limited GPU resources.”
>
> Answer (R1-Q3-A): The fine-tuning step has been implemented on a single node (1 server) with upto 4 GPUs. Compared to big AI models which require multiple nodes and several tens to hundreds of GPUs during training / fine-tuning, our fine-tuning is done with much less nodes and limited GPU resources. Please see line 395 of the revised manuscript.

---

> > ### Comment · Reviewer_4qQL · 2024-11-28
> >
> > Thank you very much for the efforts made by the authors. Honestly, the current version of the paper has finally reached the organizational standards of an academic paper. However, I regret to say that I will maintain my score, as the novelty, contribution to the community, and experimental results presented in the paper do not meet the expectations of ICLR.
> >
> > For future research, the authors might consider the following tips:
> > 1. Include more quantitative and qualitative results for reconstructed CT images, particularly covering all the target tasks discussed in the paper.
> > 2. It would be better to add comparisons with non-sinogram inpainting methods. After all, the purpose of sinogram inpainting is ultimately to achieve better CT image reconstruction. The paper does not seem to demonstrate the advantages of the inpainting approach. Does it address some critical issues inherent to image-domain methods?
> >
> > I understand that the authors have made significant efforts and emphasized that this work is focused on Synchrotron data. However, I still have some concerns. Can various methods from the medical CT field be directly applied to this type of data? Additionally, the experimental results indicate that models trained solely on phantom data perform poorly, and the amount of real data added for fine-tuning is not insignificant. These issues diminish the significance of this work.

---

### Meta-Review · Area_Chair_Ac35 · 2024-12-17

**Metareview:**

The paper proposes a foundation model for sinogram inpainting based on the latent diffusion model (LDM).  The authors incorporate physics knowledge in the loss functions, by considering certain physical characteristics of CT acquisition and sinograms. This stabilizes the training process of the autoencoder.  Experimental results demonstrate that this method outperforms current approaches in sinogram inpainting.

However, the reviewers pointed out that the structure of this paper is disorganized.  There are many different sub-modules: autoencoder, diffusion model, and blending. The loss is complex with too many parameters. The paper uses simulated projection data rather than real word data.  The proposed method only works with Parallel beam projection geometry. Training an autoencoder is hard.

In summary, all the reviewers believe that the novelty, contribution to the community, and experimental results presented in the paper do not meet the expectations of ICLR.

**Additional Comments On Reviewer Discussion:**

The reviewers believe that the overall quality of the paper still does not meet the standards of ICLR.

---

### Decision · Program_Chairs · 2025-01-22

Reject